# Developing Semi-Automated Approaches for Generating Survivorship Care Plans for Pediatric Cancer Survivors

Andrew Hornback[†], Rebecca Williamson Lewis[§], Wayne H. Liang[§], Naveen Muthu[§], Harinishree Sathu[†],
Yuanda Zhu[†], Benoit Marteau[†], Karen E. Effinger[§], May D. Wang[†]

[†] Georgia Institute of Technology, [§] Children's Hospital of Atlanta

*Abstract*—Survivorship Care Plans (SCPs) are clinical documents that summarize treatments, long-term health risks, and evidence-based recommendations for cancer and hematopoietic stem cell transplantation (HSCT) survivors. Despite their clinical value, SCPs remain underutilized due to limited automation, high documentation burden, and workflow misalignment. Manual SCP generation is time-consuming, error-prone, and burdensome—particularly in complex cases requiring hours of chart review and manual calculation. To address these challenges, we developed a semi-automated SCP generation system grounded in principles of Artificial Intelligence (AI) implementation science, focusing on clinical context-aware integration, sustainable workflow alignment, and human-centered design. The system employs an Extract, Transform, and Load (ETL) pipeline to extract survivorship-relevant data from Epic Clarity, processing both structured and unstructured Electronic Health Record (EHR) data. Structured data are processed using deterministic rules, whose outputs are reviewed by clinical experts in an iterative, human-in-the-loop process to validate accuracy and refine rule logic. Unstructured notes are analyzed using a BERT-based NLP model to identify documentation of radiation therapy. In collaboration with a large pediatric healthcare system in the United States, we retrospectively identified a cohort of patients less than age 30 treated for cancer or HSCT between January 2011 and December 2021. Using a validation cohort of 864 patients, our system achieved $\geq 99.5\%$ concordance for 53 out of 57 chemotherapy agent exposures, with most discrepancies attributable to human abstraction errors. Ongoing work includes usability testing with clinicians, co-design with survivorship coordinators, and evaluation of implementation outcomes such as trust, safety, and integration into clinical workflows.

*Index Terms*—Artificial intelligence, artificial intelligence implementation science, clinical informatics, data harmonization and integration, hematopoietic stem cell transplant, oncology, survivorship care plans

## I. Introduction

THE current estimated five-year relative survival rate for all cancers is 68%, with more than 17 million cancer survivors in the United States [1]. In pediatrics, the survival rate is even better with greater than 85% of children surviving five years from their cancer diagnosis, with more than 500,000 survivors of childhood cancer across the US [1]. The National Academy of Medicine recommends that cancer survivors participate in survivor care monitoring to help mitigate the significant physical, psychosocial, and financial consequences of cancer and its treatment [2], [3]. The basic tenets of survivor healthcare for cancer survivors are: 1) early detection of recurrence and secondary cancers, 2) management of long-term psychosocial and physical problems (i.e., late effects), and 3) preventative health care to mitigate the impact of cancer on health and quality of life. Based on current evidence in the literature and expert consensus opinion, the American Cancer Society (ACS), American Society of Clinical Oncology (ASCO), and National Comprehensive Cancer Network (NCCN) have developed Survivorship Care Guidelines to provide recommendations to assist healthcare providers in the long-term care of adult cancer survivors [4]–[9]. These diagnosis-specific guidelines list late effects survivors are at risk for, outline the surveillance and tests that are recommended, and detail the frequency at which those tests should be done. Similarly, the Children's Oncology Group (COG) has developed Long-Term Follow-Up Guidelines for Survivors of Childhood, Adolescent, and Young Adult Cancers. Unlike their adult counterparts, these guidelines use an exposure-based approach to recommend surveillance [10].

Oncology providers use these guidelines to develop survivorship care plans (SCP) for their patients comprising 1) a cancer treatment summary, 2) potential physical and psychosocial late effects, and 3) recommended evidence-based late effects surveillance [11]. SCPs are intended to facilitate the provision and coordination of high-quality survivorship care and were an integral part of the American College of Surgeons' Commission on Cancer (CoC) Guidelines in 2016. These credentialing guidelines mandated that cancer programs develop and implement processes to monitor the formation and dissemination of SCPs, which required discussion with the patient [12]. However, there were many barriers to implementation, including the effort required to create SCPs. Therefore, the updated standards in 2020 no longer required distribution of SCPs, and instead focused on establishing a survivorship program with encouragement to provide SCPs to patients [12], [13].

While it is difficult to gauge the true prevalence, studies have consistently shown low SCP utilization in adult oncology [14]–[17]. SCPs are time-consuming and expensive to create and there is no consistent method of delivery [14], [16], [18]. Several reviews have examined outcomes associated with SCP delivery and have found mixed results with limited benefits [15]–[18]. In studies of primary care providers, there is an increase in survivorship conversations and knowledge [15].

However, studies focused on survivor outcomes have found mixed results on the increase in recommended screening or preventive care [15]–[18].

Unlike in adult oncology, the use of SCPs in pediatric oncology has increased greatly over time. In a survey of COG institutions, 88% of responding institutions reported delivery of a cancer treatment summary in 2017 compared to 67% in 2007 [19]. In this population, there have been limited studies about the effectiveness of SCPs on outcomes. In a retrospective study at our collaborating pediatric center, it was found that 56% of 3,394 eligible survivors diagnosed between 2002-2016 received a SCP at a median of 0.7 years from eligibility to survivor clinic services. In a multivariate model, SCP receipt was associated with improved survival (aHR 0.31, 95% CI: 0.31-0.46) [20].

While adoption of SCPs in pediatric care has increased and have shown some potential to improve survival, the time and cost to create SCPs remains problematic. When pediatric programs were surveyed about their primary barriers to survivor care, lack of dedicated time (58%) and not enough funding (41%) were the two largest barriers. For survivors of pediatric cancer, SCPs are crucial as often these patients are too young to remember their treatment and will eventually transition to adult healthcare [19]. Currently, no tools exist for automatically generating SCPs from Electronic Health Record (EHR) data and standard EHR tools do not adequately support SCP generation. For example, determining lifetime doses of chemotherapy is critical for determining SCP recommendations, but the existing lifetime dose calculator tool in Epic Systems, the most widely used EHR system in the U.S., with adoption across 45% of all hospitals and 64% of children's hospitals [21], is unreliable as it does not take into account complex age, weight and body surface area (BSA) dosing rules and conversions in pediatrics. Passport for Care (PFC) is the closest tool available, but it requires manual data entry to generate SCPs with no ability to directly access and use EHR data [22].

To address persistent barriers to SCP creation, we have designed a novel, modular software pipeline informed by principles of Artificial Intelligence (AI) Implementation Science. Our system employs a human-in-the-loop, semi-automated workflow that leverages commonly available EHR data fields to support real-world integration of AI-enabled SCP generation. Designed for future deployment in a large pediatric health system with a high-volume oncology program, the pipeline uses an Extract, Transform, and Load (ETL) framework to process data from the Epic Clarity clinical data warehouse [23], extracting and transforming key survivorship-relevant elements, including antineoplastic medication history and cumulative dosages.

The main contributions of this work focus on implementation readiness, addressing key determinants of future success such as technical feasibility, system adaptability, and contextual fit:

- We introduce an ETL framework that cleans, harmonizes, and standardizes raw EHR data needed for SCP gen-

eration, including normalization of dosage units, drug identifiers, and structured clinical events. In addition to chemotherapy exposure and cumulative dosing, our system calculates a comprehensive range of survivorship-relevant data elements, including:
  – Individual antineoplastic drug exposures and dosages
  – Radiation therapy exposure
  – Exposure to radiopharmaceutical therapies (e.g., metaiodobenzylguanidine (MIBG), Iodine-131 (I-131))
- We emphasize a modular architecture that supports integration into broader clinical informatics ecosystems without requiring extensive code refactoring or platform-specific dependencies. The system was developed entirely using data fields standard to Epic; as a result, it is engineered for scalability, transportability, and cross-institutional dissemination to the majority of cancer treatment centers in the U.S. Extension to non-Epic systems (e.g., Oracle Health) would involve targeted mapping of source data elements but not a full system re-engineering, preserving adaptability across healthcare settings. To our knowledge, we are the first to develop a pipeline that semi-automates the generation of SCPs using raw EHR data.

## II. METHODOLOGY

To enable automated generation of SCPs, we designed and implemented a modular ETL pipeline that integrates structured and unstructured data from multiple clinical sources. The pipeline extracts survivorship-relevant variables such as radiation exposure, antineoplastic drug histories, and radiopharmaceutical treatments, transforming raw clinical data into computable elements that support SCP creation. Our approach balances algorithmic performance with clinical interpretability and fidelity to real-world workflows.

Figure 1 summarizes the core data sources, extraction methods, and resulting outputs. The pipeline draws from three primary sources: (1) patient imaging history files, (2) Epic Clarity patient records, and (3) unstructured clinical notes. Each data source is processed using a domain-specific method, ranging from Python-based file parsing to deep learning–based natural language processing (NLP). Extracted outputs include binary indicators of radiopharmaceutical therapies (e.g., MIBG and I-131), detailed drug exposure histories with dosages, and binary or probabilistic assessments of radiation exposure inferred from clinical documentation.

This modular architecture allows for parallel development and validation of each component, ensuring the system can be iteratively improved and adapted to evolving clinical documentation practices. In the following sections, we describe each data extraction module in detail.

### A. Drug Exposures and Dosages

The ETL process for drug exposures and dosages integrates three Epic Clarity datasets: Medication Orders (MO), Medication Administration Records (MAR), and Patient Height and

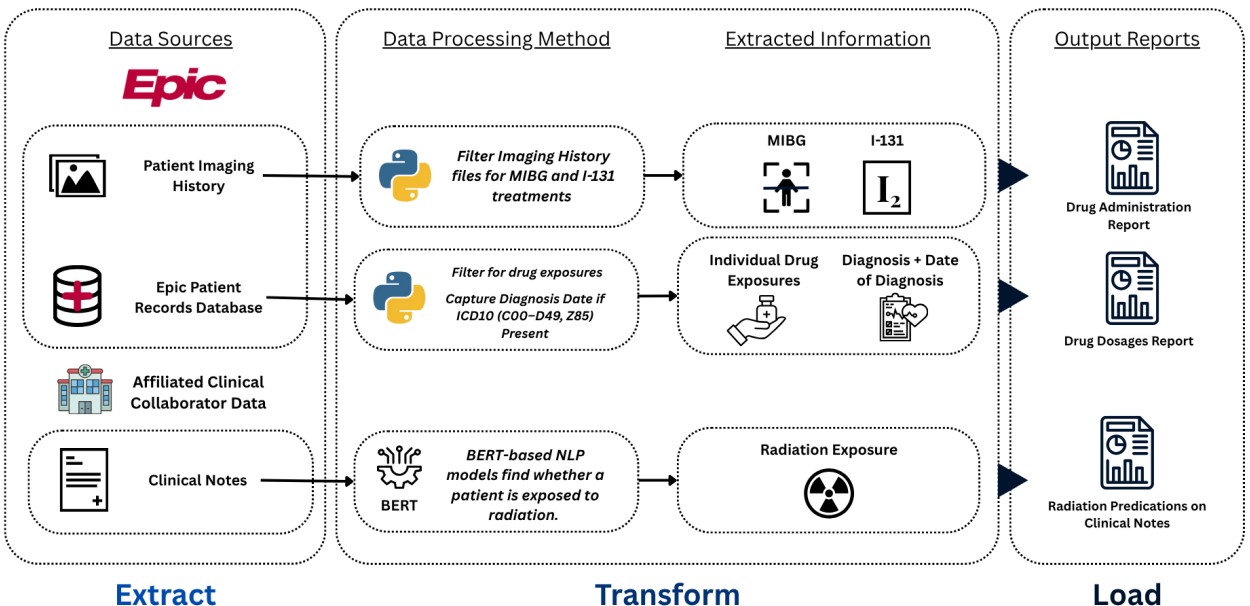

Fig. 1. Overview of the Extract, Transform, and Load (ETL) pipeline. The data sources used are from Epic, which consisted of patient imaging history and patient records, and data from our clinical collaborator, which had clinical notes. The patient medical imaging history was filtered for presence of Iobenguane (MIBG), a radiopharmaceutical used for diagnosis, and Iodine-131 (I-131), a nuclear cancer treatment; this was concatenated into a drug administration report. The Epic patient records database filtered for individual drug exposures and separately, captured diagnosis type and date if relevant ICD-10 codes were present. These attributes were combined into a drug dosages report. For the clinical collaborator notes, we utilized BERT-based NLP models to extract presence of radiation treatment, which was then added to a separate Radiation Predications report. This modular design supports portability across institutions, and data processing decisions are clearly documented and reproducible.

Weight History (PHWH). These are pulled via SQL scripts and fed into the pipeline.

MO includes ordered medications along with patient identifiers (ID, name, Date of Birth (DOB)) and physical data such as height, weight, and BSA. MAR contains similar identifiers and physical measurements, plus administration details such as route (e.g., intravenous, intrathecal), administration time, dose, and dose units (e.g., milligram, milligram per kilogram (mg/kg), milligram per meter squared (mg/m$^2$)). PHWH logs historical height (inches) and weight (ounces).

The ETL process consists of eight steps: preprocessing height and weight, determining medication order frequencies, mapping MAR medications to generic names, counting MAR orders, aggregating dosages at the order and patient levels, exporting to a standardized validation format, and validating exposures and dosages.

*1) Preprocessing Height and Weight:* In pediatrics, due to varying patient sizes, antineoplastic drugs are typically dosed by BSA or weight (e.g., mg/m$^2$, mg/kg). Accurate cumulative dosing for SCPs relies on the MO and MAR datasets, but MO entries often lack BSA or the height and weight needed to compute it—an issue that contributes to errors in both the Epic lifetime dose calculator and manual abstraction. To address this, we use the PHWH dataset to fill in missing BSA or weight values and improve dosage accuracy. Height and weight are recorded in inches and ounces and must be converted to centimeters and kilograms, respectively, to calculate BSA via the Mosteller Formula:

$$\text{BSA} = \sqrt{\frac{\text{Height} \times \text{Weight}}{3600}} \quad (1)$$

After conversion, MO data is passed to a function that finds the nearest height and weight entries from PHWH based on the MO's Ordering Date and PHWH's Recorded Time, one metric at a time for efficiency. The two values need not be from the same date but must be within 30 days of the order, as pediatric growth can make older values unreliable. Once both values are obtained, BSA is calculated using Equation 1 and substituted into the MO record as Height at Release, Weight at Release, and BSA at Release.

*2) Establishing Medication Order Frequencies:* Each medication order in the MO file corresponds to an administration record in the MAR file for on-site medications. Orders for at-home administration typically have no matching MAR entries. Each MO record includes a frequency value, indicating the maximum number of doses allowed within a time frame—for example, 5 TIMES A DAY implies up to 5 doses per day, while Q6H allows up to 4 (i.e., one every six hours).

A mismatch can occur when the actual number of MAR entries exceeds the order frequency—e.g., a once-daily medi-

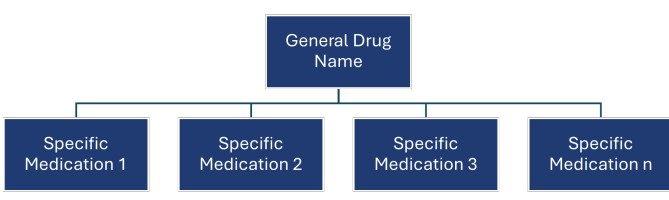

Fig. 2. Mapping specific chemotherapy medications (distinguished by dose, route, etc.) to their general drug names.

cation may appear twice in the MAR. This usually results from dose splitting based on clinical judgment or re-administration after a failed dose (e.g., emesis after oral chemotherapy). Though the total dose remains unchanged, the MAR shows multiple entries.

To address this, we mapped each possible frequency value in the MO to a numeric upper limit on daily administrations, which is then used to guide dosage calculations.

*3) Mapping Medications in the MAR to General Drug Names:* Chemotherapy drugs requiring lifetime cumulative dosage calculations (per BSA or weight) fall into four classes: anthracyclines, alkylators, heavy metals, and other agents.

Each general drug name corresponds to multiple specific medications. To accurately calculate lifetime dosages, we mapped specific medications to their general drug names through a two-step process: first, pattern matching general drug names within specific medication names in the MO data (which often include dosage and administration details); second, manual mapping of exceptions. Figure 2 illustrates this process, with doxorubicin mappings detailed in Table I.

TABLE I
EXAMPLE OF A SPECIFIC MEDICATION MAPPING TO A GENERAL DRUG NAME

| General Drug Name | Specific Medication |
|---|---|
| doxorubicin | doxorubicin 20 mg/10 ml intravenous solution |
| doxorubicin | doxorubicin pegylated liposomal (doxil) iv solution |
| doxorubicin | doxorubicin 2 mg/ml intravenous solution wrapper |
| doxorubicin | doxorubicin iv solution |

*4) Creating Record Counts for Each Order in the MAR Data:* The MAR dataset records administrations for orders listed in the MO file for on-site medications. Originally, each MO order has a single record with dosage and frequency data. To harmonize inpatient data, dosages are calculated from a modified MO file that expands each order into multiple records reflecting the number of actual administrations.

This pipeline step maps each order ID to the count of corresponding MAR records. The count is then checked against frequency limits established earlier. Separating these steps reduces computational complexity compared to analyzing MO and MAR files simultaneously.

After validating this mapping, the MO data is adjusted by duplicating records to match MAR counts. This produces a harmonized MO dataset with accurate records for both inpatient and outpatient administrations, enabling integrated chemotherapy dosage calculations.

*5) Aggregating Medication Dosages at the Order Level:* Once the MO data are harmonized, for each order ID and medication pairing, the total dosages are calculated. Calculating grouped order ID and medications prior to total per patient dosing reduces latency in the pipeline and provides a mechanism for quality control, as the data can be checked in Epic Clarity at the order ID level, enabling debugging without having to search through a patient's entire history.

*6) Aggregating Medication Dosages at the Patient Level:* Aggregating dosages from the order to patient level begins by converting doses to milligrams per meter squared ($mg/m^2$). For example, doses in mg/kg are multiplied by 30 to convert to $mg/m^2$ according to standard clinical practice, while doses in milligrams are divided by the dosing BSA at administration, accounting for BSA changes over time.

Besides unit conversion, minor rules address combination drugs. For instance, the medication *Daunorubicin 44 mg and Cytarabine 100 mg in liposome IV solution* is a mixture containing two drug components. In this case, dosages are split by extracting general drug names via regular expressions and applying transformation rules, yielding two records instead of one.

The stage outputs a table with Patient ID, Medication, and $mg/m^2$ Dose. Sample synthetic data from this step is shown in Table II, where specific medications are aggregated at the patient level before general drug name aggregation in the next stage.

TABLE II
SAMPLE OUTPUT FROM AGGREGATING MEDICATION DOSAGES AT THE PATIENT LEVEL ILLUSTRATING HOW CUMULATIVE ANTINEOPLASTIC DRUG EXPOSURES ARE COMPUTED BY THE ETL PIPELINE.

| Patient ID | Medication | $mg/m^2$ Dose |
|---|---|---|
| 100 | Doxorubicin 10mg/5ml intravenous solution | 255.00 |
| 101 | Daunorubicin 5mg/ml intravenous solution | 300.00 |
| 102 | Busulfan 60 mg/10ml intravenous solution | 266.78 |

*7) Converting the Output to a Standard File Format used to Validate Against Internal Data by our Collaborator:* In this stage, patient drug exposures and aggregated dosages are converted into a collaborator-defined file format for validation. Each drug is represented by a binary variable indicating exposure, alongside the cumulative lifetime dose calculated previously. This step ensures full integration with the collaborator's established format for manual data inspection.

*8) Validating the Exposures and Dosages:* The final pipeline step performs quality control by comparing binary exposure variables and doses against a supposed ground truth validation data file from a manually curated clinical registry. While validation is conducted with clinical collaborators, this stage improves the informatics pipeline by displaying current values, previous values, and the ground truth to monitor

changes and ensure the intended effects of pipeline modifications.

## B. Cancer Diagnoses and Diagnoses Dates

To identify cancer diagnoses, we queried the Epic problem list file for active and resolved entries containing ICD-10 codes in the ranges C00–D49, which encompass malignant neoplasms and other neoplastic conditions, and Z85, which denotes a personal history of malignant neoplasm. For each identified record, we extracted both the diagnosis code and the associated diagnosis date. This allowed us to capture active and historical cancer diagnoses as well as the temporal context necessary for downstream survivorship care plan generation and clinical validation.

## C. MIBG and I-131 Therapy

To identify patients who received I-131 or MIBG therapy, we analyzed the institutional imaging history data. For I-131 exposure, we filtered for imaging procedures labeled "NM THYROID ABLATION", which corresponds to nuclear medicine–based thyroid ablation procedures typically involving I-131. For MIBG, we extracted records with procedure descriptions of either "RADIOTHERAPY" or "NON-THYROID", both of which encompass therapeutic MIBG scans administered for neuroblastoma and other relevant indications. This approach enabled structured extraction of exposure data for these radiopharmaceuticals from unstructured imaging clinical reports, providing a reliable method to detect and validate patient-level treatment history.

## D. Radiation Therapy Exposure

To classify radiation exposure from clinical notes, we developed a binary classification model using Bidirectional Encoder Representation from Transformers (BERT) models [24]. We manually labeled 1,500 clinical notes from 76 patients, identifying whether each note indicated evidence of radiation exposure. These labeled data were split into training and test sets using an 80/20 ratio. Standard text preprocessing steps were applied, including lowercasing and tokenization using the 'BertTokenizer' from the pretrained 'bert-base-uncased' model. Fine-tuning was performed over 50 epochs with a batch size of 16 for both training and evaluation. The model was optimized to identify mentions of radiation exposure at the clinical note level, enabling patient-level classification when aggregated. We used base BERT rather than domain-specific variants (e.g., ClinicalBERT, BioBERT) due to its strong baseline performance, broad generalization ability across varied clinical note formats, and compatibility with our local fine-tuning corpus. Given the diversity of radiation documentation in our institution's EHR, base BERT offered a flexible and effective solution without requiring domain-specific pretraining.

## III. RESULTS

We correctly validated the specific cancer diagnosis for 756 patients (87.5%), an insufficient specific cancer diagnosis in 72 patients (8.3%), no cancer diagnosis for 20 patients (2.6%),

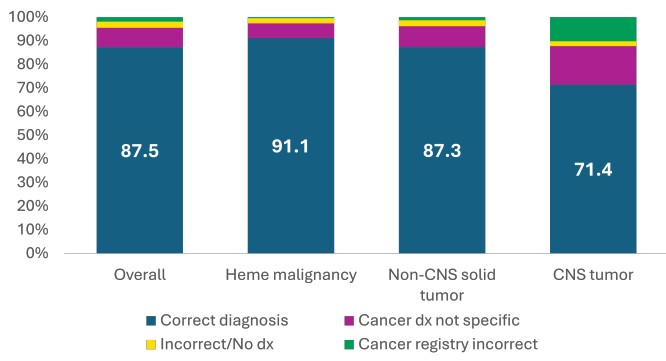

Fig. 3. Proportion of patients with correct diagnoses (dark blue) across four cancer categories based on comparison with manual chart review. The ETL pipeline achieved highest concordance for hematologic malignancies (91.1%) and lowest for CNS tumors (71.4%), with remaining discrepancies due to nonspecific documentation, missing diagnosis codes, or cancer registry mismatches.

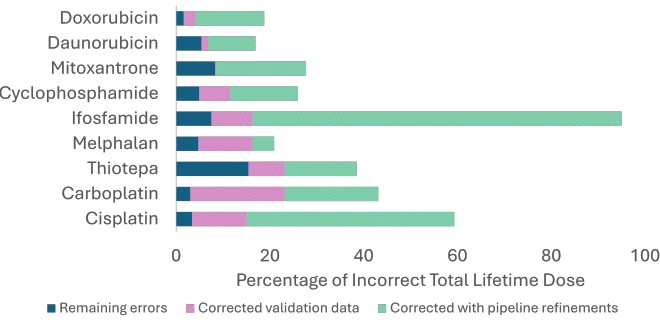

Fig. 4. Proportion of errors in total lifetime dose (mg/m²) by drug type, showing remaining errors and those corrected through validation and pipeline refinements (errors >5% discrepancy).

and found a diagnosis more correct than the cancer registry validation dataset for 16 patients (1.9%), as shown in Fig. 3, date of diagnosis was correct within 30 days for 89.8% of patients.

Our ETL pipeline demonstrated high accuracy in identifying patient chemotherapy exposures, correctly matching ≥99.5% of exposures for 53 out of 57 chemotherapy agents. Notable agents with slightly lower match rates included thioguanine (99.2%), dinutuximab (98.6%), anti-thymocyte globulin (ATG, 98.4%), and sirolimus (98.0%). Among the 100 total discrepancies identified, 45% were due to human abstraction errors in the validation data detected by our pipeline, 21% were related to treatment administered at outside institutions, 16% were exposures that were not cancer-related, 15% were exposures missed by the pipeline, 2% were prescriptions that were never administered, and 1% were intracavitary administrations not captured in structured data.

For cumulative dosage validation, initial error rates (defined as discrepancies greater than 5% from the validation data) for anthracyclines ($n = 4$), alkylating agents ($n = 9$), and heavy metals ($n = 2$) were 29.6%. Following refinement

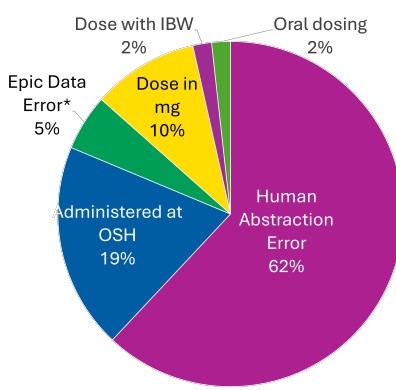

Fig. 5. Reasons for Total Lifetime Dose Discrepancies >5%. IBW: Ideal Body Weight, OSH: Outside Hospital. Data errors include: Missing administrations, partial dosing orders, IV medications ordered through home health.

of the pipeline and correction of errors in the validation dataset uncovered by our pipeline, the discrepancy rate was reduced to 6.6%, as illustrated in Figure 4. Most of the original discrepancies were due to either human abstraction error (62%) in the validation data or administration at and outside hospital (19%), as shown in Fig. 5.

In radiopharmaceuticals, our pipeline identified one discordant I-131 exposure due to administration after the manual abstraction date. All seven patients who received MIBG treatment at our institution were correctly detected. The NLP model for radiation exposure achieved a 98% AUROC with a 0.009% false positive rate and zero false negatives at the note level. While some notes were misclassified, every patient with radiation had at least one correctly identified note—meaning a patient was labeled positive if any of their notes were true positives.

## IV. DISCUSSION

SCPs have shown benefits in pediatric oncology [20] and are increasingly adopted [19]. To accelerate adoption and reduce provider burden, SCP creation must be faster, minimize errors, and enable quicker information delivery, which is especially critical for smaller programs with limited resources. Manual abstraction errors must also be eliminated.

In our pipeline evaluation, 45% of treatment exposure discrepancies were due to errors in external validation data that our pipeline correctly identified, underscoring the need for more accurate, semi-automated processes to improve clinical workflows and patient outcomes. Incorrect exposures and dosages risk inappropriate surveillance or education, potentially delaying interventions or causing costly over-screening.

Additionally, 21% of exposure discrepancies stemmed from administration at outside institutions, highlighting the necessity for manual review alongside automation for multi-institution patients. Integrating external treatment data is challenging; data often arrives in unstructured formats like PDFs rather than interoperable standards-based files and are not integrated into the corresponding EHR data fields containing locally administered treatment details. To address this, we plan to develop NLP models to detect external treatments from clinical notes and flag entries prone to errors (e.g., certain combination drugs, mg dosing in young patients) for manual review, adding a human-in-the-loop component.

There are also notable challenges we currently face. One of the primary obstacles is data interoperability and standardization. EHR systems, like Epic, store heterogeneous data—ranging from structured medication records to unstructured clinical notes—often in siloed databases. Extracting, transforming, and harmonizing this data for AI use, particularly in NLP applications, demands significant engineering effort. This challenge is amplified in pediatric oncology, where longitudinal care records include varied treatment modalities and long-term follow-up data that may span multiple institutions and clinical domains.

Another core challenge is model reliability and clinician trust. Even when AI tools like BERT-based NLP systems achieve high performance, their adoption in clinical workflows depends on transparency, explainability, and alignment with decision-making. Clinicians are often hesitant to rely on opaque models, particularly for tasks with direct patient-care implications such as generating SCPs. To foster trust, models must not only perform accurately but also present outputs in ways that clinicians can interpret, verify, and adjust as needed.

Operationally, scaling AI solutions across large hospital systems requires institutional buy-in, governance frameworks, and ongoing maintenance. This includes IT support for continuous data pipeline operation, mechanisms for monitoring model drift and retraining, and compliance with institutional privacy and security policies. AI tools must also accommodate differences in workflows across specialties and clinical teams, which can necessitate configurable, modular designs rather than one-size-fits-all deployments. Our pipeline currently runs on a 32GB RAM, 4-core CPU Red Hat server and takes approximately 30 minutes to process our sample institutional cohort. To improve scalability and performance, we are migrating to a cloud-based system with 64GB RAM and 16 CPU cores. Trained model weights and pipeline code are stored and version-controlled on GitHub to support reproducibility, ongoing updates, and maintenance.

Finally, regulatory and ethical considerations, such as compliance with healthcare laws, data provenance, and patient consent, must be carefully addressed when deploying AI in clinical settings. These factors are particularly salient when working with vulnerable populations like pediatric cancer survivors, whose data may be subject to additional privacy protections and ethical oversight.

## V. CONCLUSIONS

Large-scale clinical informatics systems that operate on EHR data in major healthcare organizations often face substantial data engineering challenges, regardless of the specific application [25]. In the domain of SCP generation, the only currently available system, PFC, remains limited as it

lacks EHR integration, requiring manual chart review and data entry outside the EHR. In a PFC study, 43% of users reported that documenting a simple case of uncomplicated acute lymphoblastic leukemia treatment required more than 30 minutes for medical record abstraction and data entry, with more complex cases increasing this burden substantially [26].

To our knowledge, we are the first to develop a semi-automated SCP generation system in a real-world clinical setting aimed at improving pediatric cancer survivorship care and team efficiency. Grounded in AI implementation science, our work emphasizes not only a technically sound solution but also workflow integration, frontline usability, and scalability. Implemented within a large pediatric oncology program, our pipeline extracts SCP data elements from a widely used EHR, potentially reducing documentation time while validating chemotherapy drug calculations against manual methods. It addresses key implementation challenges by (1) integrating with EHR data to minimize workflow disruption and support decision-making, (2) enabling SCP data use in research to promote continuous learning, and (3) incorporating pediatric-specific factors such as age-, weight-, and BSA-based dosing—features lacking in current EHR solutions. Designed for interoperability and sustainability, our system is compatible with Epic Clarity and adaptable to other EHRs via a common data model.

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
