# OpenReview forum: "Developing Semi-Automated Approaches for Generating Survivorship Care Plans for Pediatric Cancer Survivors"
_IEEE.org/EMBS/BHI/2025/Conference — BHI 2025_

### Official Review · Reviewer_6Dog · 2025-07-11
**This paper introduces a transformative semi-automated system for generating Survivorship Care Plans directly from EHR data, addressing a critical bottleneck in pediatric oncology care. The robust ETL pipeline achieves exceptional accuracy in extracting complex treatment histories while identifying errors in human-validated datasets. Grounded in AI implementation science, this work represents a significant leap toward scalable survivorship care delivery.**

**Confidence:** 5
**Clarity Of Writing:** great
**Clinical Significance:** good
**Methodological Novelty:** good
**Overall Rating:** 8

**Experiments And Results:**

good

**Questions For The Authors:**

How does the system handle edge cases, missing data, or conflicting information? What safeguards prevent propagation of errors?

**Strengths:**

The paper directly addresses a critical clinical need by automating SCP generation to reduce provider burden, particularly vital for pediatric survivors transitioning to adult care. The system demonstrates exceptional technical robustness with its modular ETL pipeline achieving high accuracy rates while handling complex pediatric-specific dosing calculations that current EHR tools cannot manage. The rigorous validation methodology impressively uncovered substantial human errors in the "ground truth" dataset, highlighting the system's potential to enhance data quality beyond mere automation.

**Summary Of The Paper:**

This study develops a semi-automated computational system for generating Survivorship Care Plans (SCPs) for pediatric cancer survivors, addressing the critical barrier of manual creation burden. The system employs an ETL pipeline to extract and harmonize survivorship-relevant data from Epic Clarity, computing patient-specific antineoplastic treatment histories and cumulative drug exposures while accommodating complex pediatric-specific dosing rules. Validated on 864 patients, the system achieved ≥99.5% concordance for 53 out of 57 chemotherapy agents and demonstrated superior accuracy by identifying human abstraction errors in validation data.

**Weaknesses:**

The validation is limited to a single institution, restricting evidence of generalizability across diverse EHR systems and documentation practices. The system incompletely handles external treatment data, requiring manual review for treatments administered at outside institutions, which is a significant limitation affecting full automation for complex pediatric patients. The paper lacks crucial implementation details including computational requirements, processing times, and model maintenance strategies, while missing empirical evidence from usability testing with clinicians.

---

### Official Review · Reviewer_jx19 · 2025-07-18
**Developing Semi-Automated Approaches for Generating Survivorship Care Plans for Pediatric Cancer Survivors**

**Confidence:** 4
**Clarity Of Writing:** great
**Clinical Significance:** great
**Methodological Novelty:** great
**Overall Rating:** 7

**Experiments And Results:**

great

**Questions For The Authors:**

1.	In the abstract, it is mentioned that AI, but what kind of algorithms, models, or NLP techniques are used in the system should be specified.
2.	This paper mentions “AI tools like BERT-based NLP systems
achieve high technical performance.”  AUROC is achieved with the highest result, which should clear some performance metrics results. (e.g., precision/recall metrics at the patient level.) It is a little bit confusing.
3.	 In Table II, the sample output only shows three patients who would be reliable for the verification, which should be mentioned clearly in this table.

**Strengths:**

The research problem is interesting, and it would be helpful in the healthcare domain. The validation is strong for the identified specific cancer diagnoses.

**Summary Of The Paper:**

This paper primarily focuses on the system for generating computational Survivorship Care Plans, which utilizes an Extract, Transform, Load (ETL) pipeline to automate the extraction of survivorship-relevant data from Epic Clarity. That pipeline process structure and unstructured Electronic Health Record (EHR). However, this work is well written and readable.

**Weaknesses:**

Figures 3 and 4should be written in a detailed caption, and check grammatical errors. “ Freq Name” makes corrections. Updated the recent references for the last five years.

---

### Official Review · Reviewer_cZrt · 2025-07-20
**Review of Paper #332**

**Confidence:** 3
**Clarity Of Writing:** excellent
**Clinical Significance:** excellent
**Methodological Novelty:** great
**Overall Rating:** 7

**Experiments And Results:**

great

**Questions For The Authors:**

Please see my comments above.

**Strengths:**

- The paper is well written and organized.
- The authors have described clearly the limitations of existing studies and have stated how the present study differs from existing ones.

**Summary Of The Paper:**

This paper presents a semiautomated Survivorship Care Plan (SCP) generation system in a real-world clinical setting aimed at improving pediatric cancer survivorship care and enhancing treatment team efficiency.

**Weaknesses:**

No specific limitation.

---

### Official Review · Reviewer_ieUb · 2025-07-20
**Development of an automated pipeline for generating pediatric survivorship care plans**

**Confidence:** 3
**Clarity Of Writing:** excellent
**Clinical Significance:** fair
**Methodological Novelty:** poor
**Overall Rating:** 3
**Final Rating:** 7

**Experiments And Results:**

fair

**Questions For The Authors:**

As stated in the weaknesses section, I would like to understand if I missed the relevant parts of the work which addressed the human-in-the-loop and usability aspects. I would also like to see an example of the final reports generated, perhaps as a supplementary screenshot, since the authors mention usability as one of the goals of the project.

**Strengths:**

- The paper describes a novel approach to solve an unmet clinical need, which is the automated generation of survivorship care plans.
- The authors employ a combination of techniques including rules-based processing of patient records and NLP-based binary classification to determine radiation exposure from clinical notes.
- The paper is clearly articulated.
- The work outperforms the external validation in some cases, demonstrating the value of automating certain aspects of report generation.

**Summary Of The Paper:**

This paper describes the development of an automated pipeline for generating pediatric survivorship care plans, integrating data from different modalities (imaging, patient records, and clinical notes) and generating reports informing survivorship care.

**Weaknesses:**

Perceived weaknesses listed below, to the best of my understanding:
- Where is the human-in-the-loop process? The introduction boasts this: "Our system employs a human-in-the-loop, semi-automated workflow that leverages commonly available EHR data fields to support real-world integration of AI-enabled SCP generation." Yet in the discussion, a human-in-the-loop component is described as future work. "To address this, we plan to develop NLP models... adding a human-in-the-loop component." Please either clarify which aspect of the current work employs a human-in-the-loop process, or remove the incorrect statement from the introduction.
- Where is the human-centered design? This is described as a guiding principle in the abstract, yet I did not see any illustration of an interface or a description of usability testing. Rather, this seems to also be set aside for future work.
- Absence of recommendations: my understanding is that patient recommendations are part of the survivorship care plan. Using an algorithm, with input and training from physicians, to generate appropriate recommendations would add substantial impact to the work.
- Despite a description of a sophisticated pipeline in the abstract, this appears to amount to (1) a couple of rule-based systems using fields in the Epic system, and (2) the application of an NLP classification algorithm to clinical notes to determine radiation exposure. While these add value to the existing manual process, these seem to be rather trivial computational solutions which fall significantly short of an end-to-end pipeline which could even be tested prospectively in a clinical setting.

---

### Official Review · Reviewer_Gemr · 2025-07-20
**Relevant clinical gap addressed**

**Confidence:** 4
**Clarity Of Writing:** good
**Clinical Significance:** excellent
**Methodological Novelty:** great
**Overall Rating:** 6

**Experiments And Results:**

great

**Questions For The Authors:**

1. The paper has used Epic as an example of a data source for EHRs. How would this be applicable to countries where similar EHR systems are not present?
2. Real world application of this model, along with usability testing, is yet to be seen. For example, how does this model translate to adult oncology SCPs, or other diseases?

**Strengths:**

1. Highly relevant in the clinical space and addresses a relevant gap in cancer rehabilitation.
2. Paper claims that it is the first to develop a pipeline that semi-automates the generation of SCPs using raw EHR data.
3. Methodology is unique and aims to utilize information from varied clinical sources in order to develop an SCP.

**Summary Of The Paper:**

The paper presents a semi-automated pipeline for generating Survivorship Care Plans (SCPs) for pediatric cancer survivors by extracting and harmonizing structured and unstructured EHR data using an ETL framework integrated with Epic Clarity. Using a validation cohort
 of 864 patients, the system achieved ≥99.5% concordance for 53 out of 57 chemotherapy agent exposures, with most errors coming from human abstraction errors in the validation data.

**Weaknesses:**

1. The paper states that it utilized BERT-based NLP models for extracting presence of radiation treatment. The rationale for using BERT can be further elucidated, given that there are multiple NLP models for interpreting clinical notes.
2. The practical viability of this approach, given HIPAA and other regulations, may need to be considered when deployed in real life settings.